# Extracellular vesicles from adipose-derived mesenchymal stem cells promote colony formation ability and EMT of corneal limbal epithelial cells

**Xiaoqin Li**[1], **Ryuhei Hayashi**[1,2,3*], **Tsutomu Imaizumi**[2,4], **Jodie Harrington**[2,5], **Yuji Kudo**[2,4], **Hiroshi Takayanagi**[1,6], **Koichi Baba**[1,7,8], **Kohji Nishida**[1,3*]

1 Department of Ophthalmology, Osaka University Graduate School of Medicine, Suita, Osaka, Japan,
2 Department of Stem Cells and Applied Medicine, Osaka University Graduate School of Medicine, Suita, Osaka, Japan, 3 Institute for Open and Transdisciplinary Research Initiatives, Osaka University, Suita, Osaka, Japan, 4 Basic Research Development Division, ROHTO Pharmaceutical, Ikuno-ku, Osaka, Japan, 5 Faculty of Health, Medicine and Social Care, Medical Technology Research Centre, Anglia Ruskin University, Chelmsford Campus, England, United Kingdom, 6 Research, Development and Production Department of RAYMEI Inc, Suita, Osaka, Japan, 7 Department of Advanced Device Regenerative Medicine, Osaka University Graduate School of Medicine, Suita, Osaka, Japan, 8 Visual Regenerative Medicine, Division of Health Sciences, Osaka University Graduate School of Medicine, Suita, Osaka, Japan

* ryuhei.hayashi@ophthal.med.osaka-u.ac.jp (RH); knishida@ophthal.med.osaka-u.ac.jp (KN)

## Abstract

Corneal diseases are a leading cause of visual impairment, and their treatment remains challenging. Corneal epithelial stem cells exist in the limbus, the peripheral region of the cornea, and play an important role in corneal regeneration. Here, we evaluated the effects of extracellular vesicles from human adipose-derived mesenchymal stem cells (AdMSC-EVs) on limbal epithelial cells (LECs). Colony formation assays showed that the colony-forming efficiency of LECs significantly increased in the presence of AdMSC-EVs. We next demonstrated that AdMSC-EVs accelerated the migration of LECs in a scratch assay, whereas the proliferation of LECs was decreased by AdMSC-EVs in the cell proliferation assay. RNA sequencing analysis of LECs indicated that AdMSC-EVs maintained their stem cell properties and improved epithelial-mesenchymal transition (EMT). Furthermore, after identifying the six most abundant microRNAs (miRNAs) in AdMSC-EVs, LEC transfection with miRNA mimics indicated that *miR-25*, *miR-191*, and *miR-335* were the most probable miRNA factors within AdMSC-EVs at improving colony formation ability and EMT. Taken together, our findings indicated that AdMSC-EVs enhanced the colony formation ability and EMT of LECs, and the effects of AdMSC-EVs were in-part mediated by the miRNAs within the AdMSC-EVs.

## Introduction

Stem/progenitor cells play a crucial role in organ formation and regeneration because of their potential for self-renewal and differentiation capacities, making them a promising source for

**Data availability statement:** The RNA-Seq datasets produced in this paper have been deposited at the Gene Expression Omnibus repository under accession number GSE255429 and GSE255314. The authors confirm that the data supporting the study findings are available within in the article and supplementary materials.

**Funding:** This work was supported in part by Fusion Oriented Research for Disruptive Science and Technology (JPMJFR210W) from the Japan Science and Technology Agency (JST), Osaka City Innovation Support Grant, and Grant-in-Aid for Scientific Research (23H03060 and 20H03842) from the Japan Society for the Promotion of Science (JSPS). X.L. was directly supported by Otsuka Toshimi and Hattori International Scholarships. The funders had no role in study design, data collection and analysis, decision to publish, or preparation of the manuscript.

**Competing interests:** I have read the journal's policy and the authors of this manuscript have the following competing interests: R.H. is affiliated with the endowed chair of ROHTO Pharmaceutical Co., Ltd. T.I. and Y.K. are employees of ROHTO Pharmaceutical Co., Ltd. H.T. is employed by RAYMEI Inc. All other authors declare that they do not have any competing interests.

replacing tissues affected by aging, degeneration, and other diseases [1]. The core properties of stem/progenitor cells, are believed to be governed by genetic factors, including both the presence of stem cell markers and the absence of differentiation markers [2]. In normal tissues, these properties cannot be self-sustained without a supporting microenvironment, and it requires complex cell-cell and cell-matrix interactions [3].

The cornea is a transparent tissue located at the front of the eye. It protects the eye from injury and infection, and as the primary refractive surface, it is essential for clear vision [4]. The corneal epithelium is the outermost layer of the cornea, and a self-renewing tissue continuously replenished by a population of stem/progenitor cells located at the corneal limbus, which have been reported to show colony formation ability *in vitro* [4,5]. Though stem/progenitor cells constitute a small percentage of the total cell population in tissues, they are essential for maintaining tissue homeostasis [1,3]; therefore, maintaining stem/progenitor cell properties of LECs resident in limbus is vital for regeneration of the corneal epithelium [6].

Adipose-derived mesenchymal stem cells (AdMSCs) have emerged as a pivotal and versatile resource in regenerative medicine, because of their multiple advantages, including a high yield with minimal patient discomfort, immunomodulatory properties, and multilineage differentiation capacity [7]. Exhibiting immense potential in various therapeutic applications, AdMSCs show promise for the treatment of various diseases, such as knee osteoarthritis, Crohn's disease, and acute ischemic stroke [8–10]. Based on their therapeutic properties, AdMSCs are considered a potential source for the treatment of corneal diseases [11,12]. AdMSCs have been reported to employ their therapeutic effects *via* paracrine activity, and previous studies demonstrated that conditioned medium (CM) from AdMSCs can exhibit the same therapeutic effects as AdMSCs themselves, suggesting that secreted factors are important for their function [13,14].

Extracellular vesicles (EVs) are lipid-bound particles released into the extracellular space by cells. EVs carry a variety of cargos, such as proteins, lipids, and nucleic acids. Their capacity to transfer functional biomolecules, especially microRNAs (miRNAs), to recipient cells allows them to alter gene expression and cellular activities that play roles in intercellular communication [15,16], which makes them promising candidates for therapeutic agents, from regenerative medicine to cancer and immune-related therapies [17–19]. There are three main subtypes of EVs: microvesicles, exosomes, and apoptotic bodies. Specifically, exosomes are typically 30–200 nm in diameter and are secreted by all cell types. They have been shown to participate in cell-cell communication [20,21]. As paracrine activity is a key mechanism underlying the effects of AdMSCs and EVs are one of the paracrine factors, AdMSC-EVs represent an attractive and promising area of study [22,23]. Several studies have shown that AdMSC-EVs promote tissue regeneration and accelerate wound healing [24,25]. Data on the mechanisms, however, by which AdMSC-EVs exert their effects on LECs, especially on their stem cell properties, are limited.

We investigated the effects of AdMSC-EVs on LECs. Additionally, RNA sequencing (RNA-Seq) was conducted to identify possible molecules for their function, particularly miRNAs, within AdMSC-EVs. Here, we provide findings on novel insights for the potential role of AdMSC-EVs in the treatment of corneal diseases.

## Results

### ADMSC-EVs promoted the colony-forming efficiency (CFE) of LECs

We evaluated the effects of AdMSC-EVs on LECs using colony formation assays. First, EVs from the CM of AdMSCs were separated by ultracentrifugation (Fig 1A). Scanning electron microscopy (SEM) revealed AdMSC-EVs with a rounded morphology and a size distribution of

30–200 nm, consistent with exosome-like EV subpopulations (Fig 1B). Western blotting indicated that the isolated AdMSC-EVs were enriched with putative exosome proteins, such as CD63, CD81 and TSG101, while absent of endoplasmic reticulum protein calnexin (Figs 1C and S1).

AdMSC-EVs were added to a co-culture system of LECs and seeded onto Mitomycin C-treated NIH 3T3 (MMC-3T3) feeder cells. An EV uptake assay demonstrated that AdMSC-EVs were uptaken by both LECs and MMC-3T3 feeder cells in the co-culture system (S2 Fig), and colony formation assays showed significantly increased CFE of LECs when exposed to AdMSC-EVs (EVs_30 μg/ml *vs.* Control: $p < 0.001$; EVs_120 μg/ml *vs.* Control: $p < 0.05$; Fig 1D,E).

## AdMSC-EVs enhanced colony formation and migration of LECs and decreased proliferation

To exclude the possible influence of AdMSC-EVs on MMC-3T3 cells and to assess only LECs, LECs were pretreated before co-culturing with MMC-3T3 cells to directly examine the effect of AdMSC-EVs on LECs (Fig 2A). Similarly, as during AdMSC-EV co-culture, pretreatment of LECs alone increased the CFE, as did AdMSC-EVs at a concentration 30 μg/ml (EVs_30 μg/ml *vs.* Control: $p < 0.05$; Fig 2B,C).

To assess the additional effects of AdMSC-EVs on LECs, scratch and proliferation assays were conducted to observe the cell migration and division capacities, respectively (Fig 2D). In the scratch assay, a dose-dependent acceleration of the gap closure rate, in monolayers treated with AdMSC-EVs was evident, with the concentration of AdMSC-EVs at 120 μg/ml significantly reducing the gap area after 24 h (EVs_120 μg/ml *vs.* Control: $p < 0.01$; Fig 2E,F). Additionally, the presence of AdMSC-EVs greatly decreased the proliferation of LECs in a dose-dependent manner (EVs_30 μg/ml *vs.* Control: $p < 0.01$; EVs_120 μg/ml *vs.* Control: $p < 0.001$; Fig 2G,H).

## RNA-Seq analysis of EV-treated LECs

To investigate the changes within LECs in response to AdMSC-EVs, cell samples were used for RNA-Seq analysis (Fig 3A). The heatmap of differentially expressed genes (DEGs) highlighted a similar pattern of gene expression changes within LECs exposed to AdMSC-EVs at both 30 μg/ml and 120 μg/ml (Fig 3B). The enrichment tree revealed the upregulation of processes related to cell motility, cell migration and extracellular matrix (ECM) organization (Fig 3C), which indicated pathways with the epithelial-mesenchymal transition (EMT) process. We examined the expression levels of markers related to stem/progenitor cells (*TP63*), differentiation (Cytokeratin *[KRT] 12*, *KRT13*), and EMT (*WNT5A*, vimentin [*VIM*], fibronectin 1 [*FN1*]) (Fig 3D). The expression level of *TP63* remained relatively stable between the control and EV-treated groups. However, the expression level of *KRT12* significantly decreased in the presence of AdMSC-EVs (EVs_30 μg/ml *vs.* Control: $p < 0.05$; EVs_120 μg/ml *vs.* Control: $p < 0.01$). And *KRT13* was also significantly downregulated ($p < 0.05$). Additionally, the expression levels of *WNT5A*, *VIM* and *FN1*, as well as other EMT-related genes (S2 Fig), were significantly increased upon exposure of LECs to AdMSC-EVs (*WNT5A*: $p < 0.001$; *VIM*: $p < 0.001$; *FN1*: EVs_30 μg/ml *vs.* Control: $p < 0.01$, EVs_120 μg/ml *vs.* Control: $p < 0.001$). As for morphological changes, even though cell circularity was not affected by AdMSC-EVs, cell area significantly increased at 120 μg/ml (S2 Fig).

## AdMSC-EVs could exert their effects on LECs *via* contained miRNAs

To determine the factors affecting the cellular activities of LECs, miRNAs were investigated, because they can be transferred by EVs between cells, and they have ability to regulate

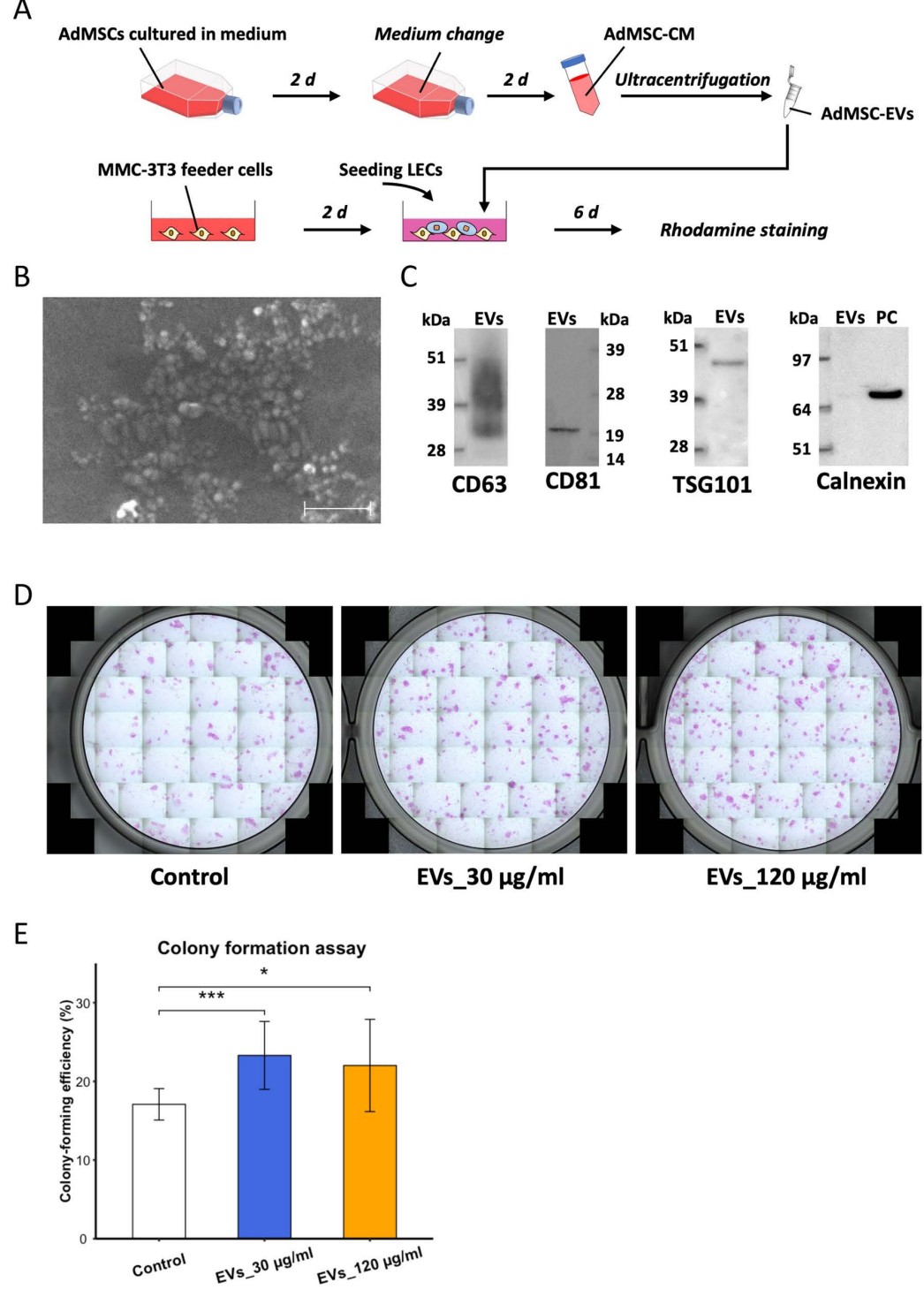

**Fig 1. AdMSC-EVs promoted the CFE of LECs.** (A) Schematic representation of the acquisition of AdMSC-EVs and establishment of colony formation assay. (B) Scanning electron microscopy of isolated AdMSC-EVs (30–200 nm). Scale bar: 0.5 μm. (C) Cropped images of CD63 (30–60 kDa), CD81 (~25 kDa), TSG101 (~46 kDa) and calnexin (~78 kDa) analysed by western blotting. AdMSCs were used as positive control (PC) for exosomal negative marker. Original blots are presented in S1 Fig. (D) Images of colonies of LECs from PBS control group and EV-treated groups (EVs_30 μg/ml and EVs_120 μg/ml). $n$ = 10 biological replicates. (E) The CFE of LECs from PBS (control) group and EV-treated groups shown as the mean ± standard deviation (SD). $n$ = 10 biological replicates. *$p < 0.05$, and ***$p < 0.001$.

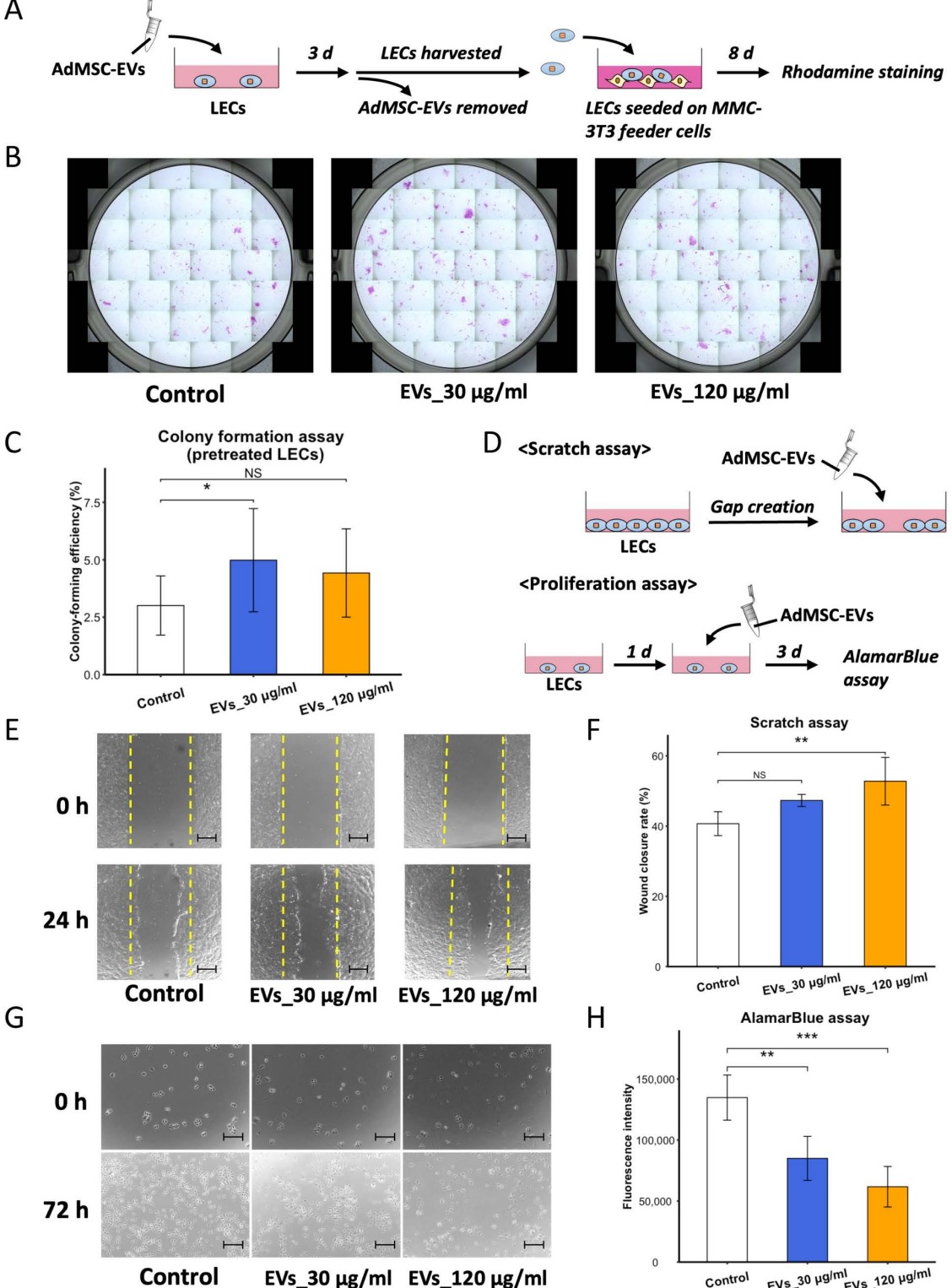

**Fig 2. AdMSC-EVs enhanced colony formation and migration of LECs and decreased proliferation.** (A) Strategy of colony formation assay using the LECs pretreated with AdMSC-EVs. (B) LECs were pretreated with either PBS (control) or AdMSC-EVs (EVs_30 µg/ml and EVs_120 µg/ml). $n$ = 12 biological replicates. (C) The CFE of LECs from control group and EV-treated groups are presented as the mean ± SD. $n$ = 12 biological replicates. (D) Schematic of scratch assay and proliferation assay. (E) Images of the control and EV-treated groups at 0 and 24 h after the gap establishment. $n$ = 4 biological replicates. Scale bar: 500 µm. (F)

Quantification of the gap closure rate of the control and EV-treated groups 24 h after gap creation. The results are shown as the mean ± SD. $n = 4$ biological replicates. (G) Images of LECs before and 72 h after treatment with AdMSC-EVs, at concentrations of EVs_30 μg/ml and EVs_120 μg/ml. Scale bar: 200 μm. $n = 4$ biological replicates. (H) Cell viability of LECs 72 h after treatment with or without AdMSC-EVs, analysed by AlamarBlue assay. The results are expressed as the mean ± SD. $n = 4$ biological replicates. NS: not significant. *$p < 0.05$, **$p < 0.01$, and ***$p < 0.001$.

expression of various genes. The top six miRNAs present in AdMSC-EVs were identified using RNA-Seq analysis (Table 1). These miRNAs were transfected into LECs to observe their individual and direct effects (Fig 4A).

The colony formation assay showed that the CFE of LECs transfected with *miR-25*, *miR-191*, and *miR-335* mimics was significantly improved compared to negative control (NC) group (*miR-25 vs.* NC: $p < 0.001$; *miR-191 vs.* NC: $p < 0.01$; *miR-335 vs.* NC: $p < 0.001$; Fig 4B,C), Based on these findings, we investigated further effects of these three miRNA mimics on LECs. At the protein level, immunolocalisation (Fig 4D) revealed minimal alteration in TP63 expression between the NC and miRNA-transfected groups. As for KRT12, the miRNA-transfected groups displayed fewer positive cells than the NC group. In contrast, compared to the NC group, miRNA-transfected groups exhibited increased numbers of VIM- and FN1-positive cells. In slight contrast to the protein-level observations, at the gene level, *deltaNP63* expression levels remained stable in the *miR-25*-transfected group and significantly increased within *miR-191-* and *miR-335*-transfected groups compared to its level in the NC group (*miR-191 vs.* NC: $p < 0.001$; *miR-335 vs.* NC: $p < 0.001$; Fig 4E). *KRT12* expression levels were less affected in the *miR-335*-transfected group, but significantly decreased in the *miR-25-* and *miR-191*-transfected groups (*miR-25 vs.* NC: $p < 0.001$; *miR-191 vs.* NC: $p < 0.001$; Fig 4E). Consistent with the immunostaining findings, expression levels of *VIM* and *FN1* significantly increased in miRNA-transfected groups ($p < 0.001$; Fig 4E). Additional markers related to stem/progenitor cells, differentiation, and EMT were evaluated (S3 Fig), and it was shown that the transfection of selected miRNA mimics resulted in the maintenance of an undifferentiated state of LECs and enhancement of the EMT phenotype. In addition, by transfecting either the three most effective miRNAs (Mix(3)) or all the highly expressed miRNAs (Mix(6)), we observed that the CFE were statistically significantly improved in both groups (S4 Fig). Gene level analysis showed that *deltaNp63* expression remained stable, whereas *KRT12* expression was downregulated in the Mix(3) group and upregulated in the Mix(6) group. As for EMT-related markers *VIM* and *FN1*, expression levels in both groups were increased compared to the NC group.

## Discussion

In this study, we showed that AdMSC-EVs enhanced the colony formation ability of LECs. Of note, RNA-Seq analysis revealed that EV-treatment improved EMT processes in LECs. More specifically, it was suggested that of the highest expressed, *miR-25*, *miR-191*, and *miR-335* were the most effective molecules within AdMSC-EVs that influenced these characteristics, highlighting the potential of AdMSC-EVs to contribute to corneal disease therapies.

AdMSC-EVs used in our study were expected to predominantly contain exosomes. Colony formation assays indicated maintenance of stem cell properties of LECs by AdMSC-EVs, which was also supported by our RNA-Seq data, showing increased cell adhesion in EV-treated LECs, as it is generally known that stem cell properties are strongly correlated with colony formation ability and cell adhesion [26]. Further investigations into the direct effects of AdMSC-EVs on LECs revealed that AdMSC-EVs promoted the migration of LECs, while concurrently decreasing their proliferation. Given that LECs remained morphologically healthy

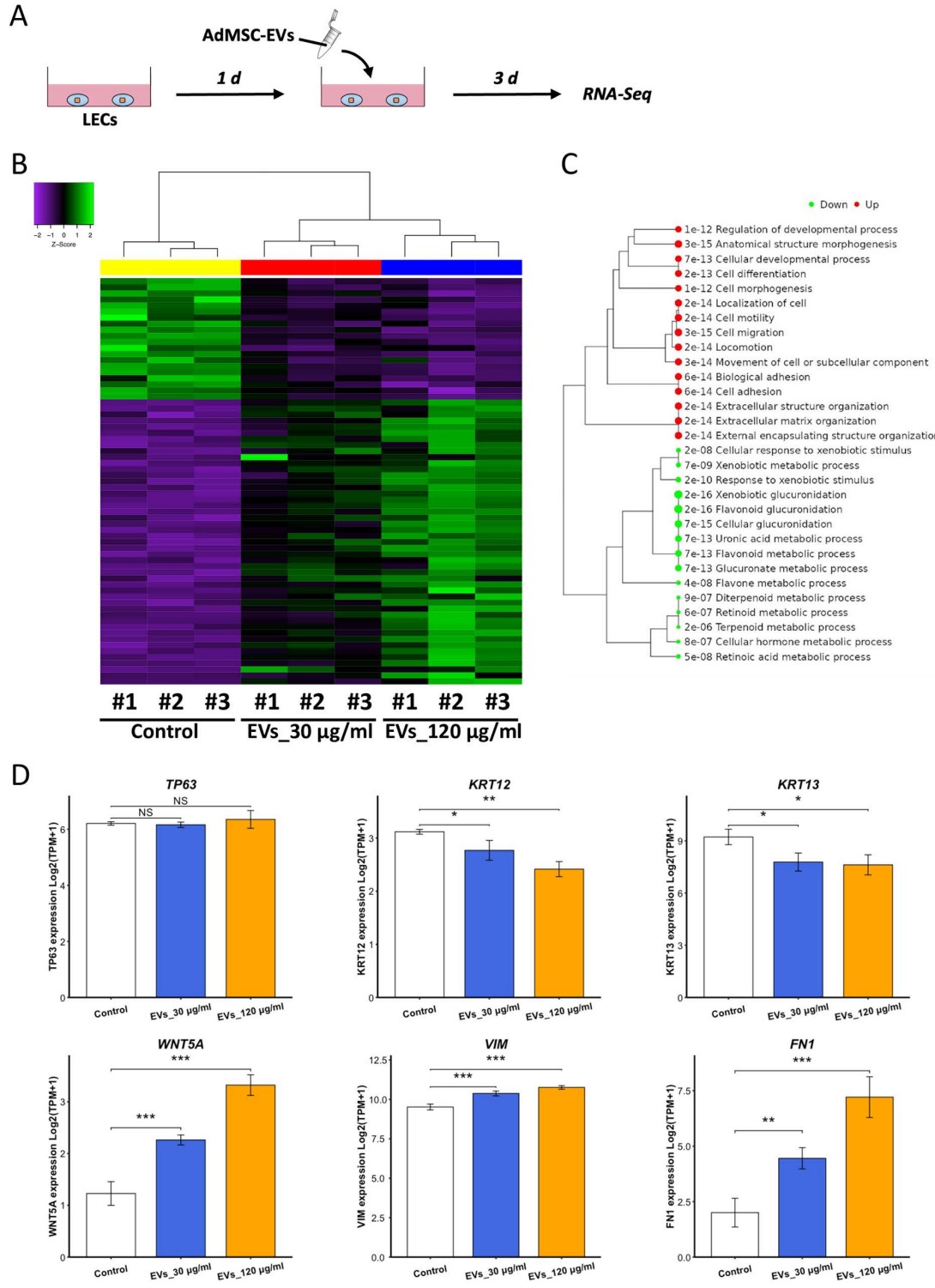

**Fig 3. RNA sequencing analysis of EV-treated LECs.** (A) Schematic of sample preparation for RNA sequencing. (B) Heatmap showing the differentially expressed genes (DEGs) of LECs from the control and EV-treated groups. $n$ = 3 biological replicates. (C) Enrichment tree sketching the upregulated and downregulated biological processes of LECs treated with AdMSC-EVs compared to the control group. $n$ = 3 biological replicates. (D) Comparisons of expression levels of genes related to stemness, differentiation, and epithelial-mesenchymal transition (EMT) between the control and EV-treated groups. $n$ = 3 biological replicates. NS: not significant. *$p < 0.05$, **$p < 0.01$, and ***$p < 0.001$.

**Table 1. The six most abundant miRNAs within AdMSC-EVs.**

| sample_1 | | sample_2 | | the top six microRNAs |
|---|---|---|---|---|
| gene_id | count | gene_id | count | |
| hsa-miR-223-3p | 5384.87 | hsa-miR-223-3p | 2749.92 | |
| hsa-miR-191-5p | 2808 | hsa-miR-335-5p | 1890.31 | |
| hsa-miR-25-3p | 2665.5 | hsa-miR-25-3p | 1772.03 | |
| hsa-miR-335-5p | 2367.5 | hsa-miR-130a-3p | 1673.17 | |
| hsa-miR-130a-3p | 2107 | hsa-miR-191-5p | 1519.25 | |
| hsa-miR-126-5p | 1694.5 | hsa-miR-146a-5p | 1344.33 | |
| hsa-miR-126-3p | 1592.5 | hsa-miR-126-5p | 1292.5 | |
| hsa-miR-93-5p | 1415.67 | hsa-miR-126-3p | 1250 | hsa-miR-25-3p |
| hsa-miR-221-3p | 1368 | hsa-miR-221-3p | 1183.83 | hsa-miR-191-5p |
| hsa-miR-23a-3p | 1348.42 | hsa-miR-23a-3p | 1027 | hsa-miR-223-3p |
| sample_3 | | sample_4 | | hsa-miR-335-5p |
| gene_id | count | gene_id | count | hsa-miR-130a-3p |
| hsa-miR-130a-3p | 3477.17 | hsa-miR-335-5p | 1869.83 | hsa-miR-126-5p |
| hsa-miR-335-5p | 2249.5 | hsa-miR-223-3p | 824.75 | |
| hsa-miR-25-3p | 1749.5 | hsa-miR-191-5p | 617.17 | |
| hsa-miR-223-3p | 1690 | hsa-miR-126-5p | 568 | |
| hsa-miR-126-3p | 1130.5 | hsa-miR-25-3p | 508.5 | |
| hsa-miR-146a-5p | 1114.75 | hsa-miR-92a-3p | 419.17 | |
| hsa-miR-126-5p | 1108.88 | hsa-miR-130a-3p | 416.33 | |
| hsa-miR-23a-3p | 1044.92 | hsa-miR-23a-3p | 379.08 | |
| hsa-miR-424-5p | 1034.92 | hsa-miR-23a-3p | 367.33 | |
| hsa-miR-221-3p | 1008.33 | hsa-miR-574-3p | 332.17 | |

during culture without floating cells in the medium (S5 Fig) and AdMSC-EVs promoted CFE of LECs, it was more likely that AdMSC-EVs decreased cell proliferation of LECs rather than inducing cell death such as apoptosis and necrosis. As AdMSCs have varied therapeutic functions depending on their microenvironment [7,27], the effects of AdMSC-EVs may also be multifaceted, owing to their interactions with different recipient cell types [28,29]. AdMSC-EVs have previously been reported to inhibit proliferation and migration of vascular smooth muscle cells *in vitro* [30] and enhance keratinocyte migration and proliferation both *in vitro* and *in vivo* [31]. In our study, AdMSC-EVs promoted migration and decreased proliferation of LECs, which might partly result from the activation of p38 MAPK signaling pathway in LECs. As previous reports have confirmed that increased p38 activity triggers accelerated cell migration, as well as suppressed ERK phosphorylation that relates to inhibition of cell proliferation [32,33]. And our analyses of the RNA-Seq data revealed increased expression levels of genes encoding p38α (MAPK14) and p38β (MAPK11), the most abundant isoforms in various tissues [34], which may indicate that AdMSC-EVs promoted migration and inhibited proliferation of LECs by activating p38 MAPK signaling pathway (S5 Fig). Additionally, in most cases, stem cells divide relatively infrequently, but they have a high proliferative potential. Contrarily, differentiated from stem cells, transient amplifying cells divide more frequently than stem cells and have a finite proliferative potential [35]; a characteristic that can be particular to primary cells sourced for regenerative medicine research [36]. In our study, the data offered an insight into the division speed of the cells but did not show an increase in cell proliferation. Based on the results, accelerated migration may have partially contributed to the higher CFE, as single LECs with increased migration ability may

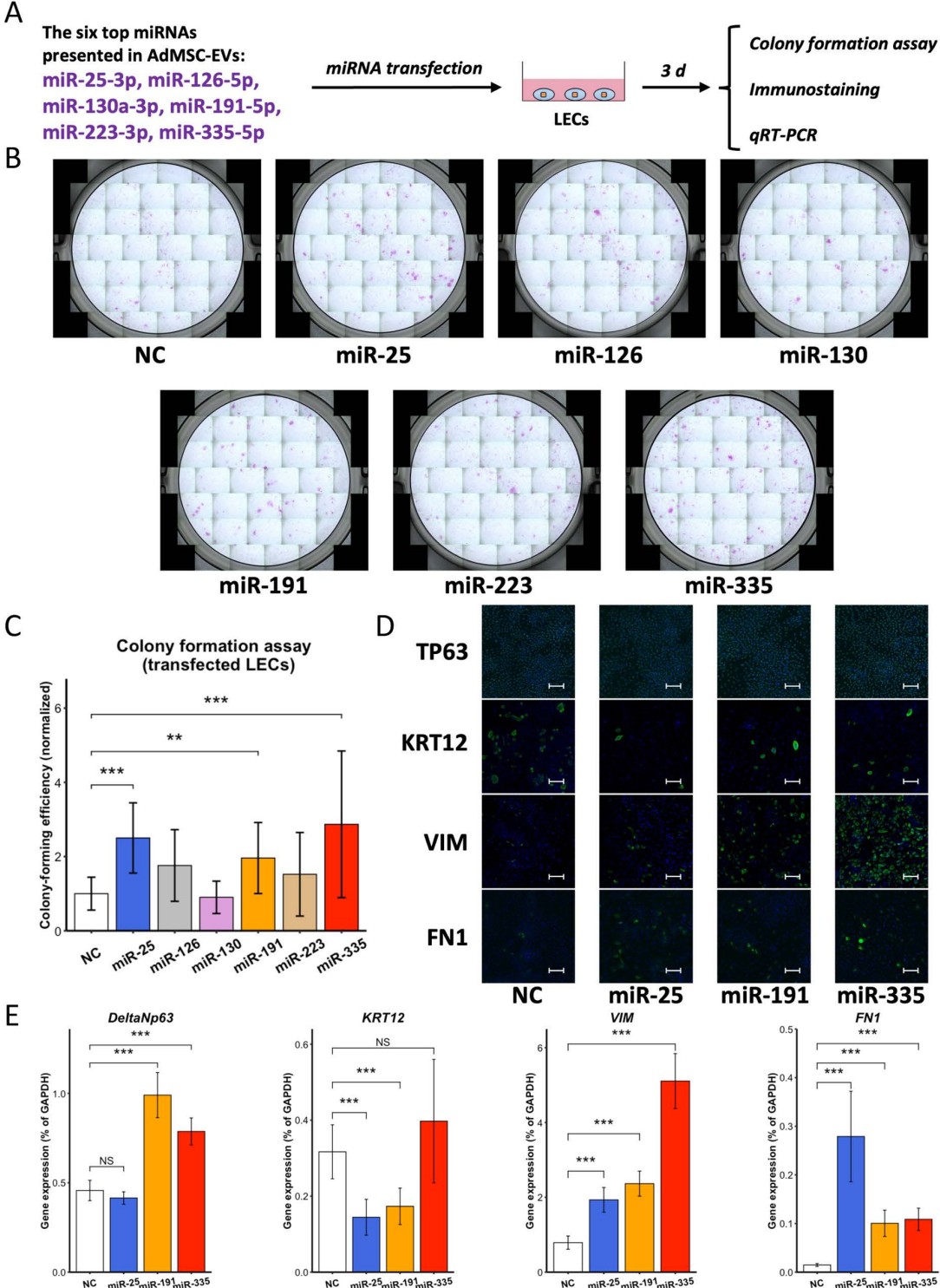

**Fig 4. AdMSC-EVs could exert their effects on LECs *via* contained miRNAs.** (A) Schematic representation of the evaluation of LECs transfected with miRNAs. (B) Images of LEC colonies from the NC and miRNA-transfected groups. $n = 16$ biological replicates. (C) The CFE of LECs from NC and miRNA-transfected groups presented as the mean ± SD. $n = 16$ biological replicates. (D) Immunostaining images of TP63, KRT12, VIM, and FN1 in LECs from the NC and miRNA-transfected groups. Scale bar: 50 μm. (E) Quantitative reverse transcription-polymerase chain reaction analysis of *TP63, KRT12, VIM,* and *FN1* between the NC and miRNA-transfected groups. $n = 8$ biological replicates. NS: not significant. **$p < 0.01$, and ***$p < 0.001$. NC: negative control.

show increased unpredictability in colony formation patterns, including increased incidences of cell reaggregation, cell splitting and colony splitting events, as reported in human induced pluripotent stem cells and human embryonic stem cells [37,38].

EMT is a biological process associated with wound healing and tissue regeneration [39]. Previous studies have confirmed that upregulated EMT in corneal epithelial cells facilitates the corneal epithelial wound healing process [40,41]. EMT initiates as a repair-associated event ensuing trauma or inflammatory injury. During EMT, polarised epithelial cells undergo multiple biochemical changes, including improved migratory ability and increased production of ECM components, in assuming a mesenchymal cell phenotype. Hence, the functional transition of polarised epithelial cells into mobile mesenchymal cells that secrete ECM components is an essential hallmark of EMT [39,42]. Our RNA-Seq analysis revealed that EMT was promoted in LECs. This is also supported by the observation that AdMSC-EVs accelerated the migration of LECs. Therefore, the individual gene expression analyses focused on markers related to stem cell properties, differentiation, and EMT. *TP63,* especially the *deltaNp63* isoform is a well-established stem cell marker of LECs, as the corneal epithelium is regenerated by *TP63*-expressing LECs under normal conditions [43]. *KRT12* is a unique biomarker for terminally differentiated corneal epithelium, and is absent from other stratified epithelial cells, such as conjunctival, oral mucosal, or skin epithelial cells [44]. *KRT13* expression is specific to non-corneal epithelial cells, specifically conjunctival epithelial cells of healthy ocular surfaces, making it a known differentiation marker [45]. Our data suggested that AdMSC-EVs contributed to maintaining the stem cell properties of LECs by inhibiting their differentiation. *WNT5A*, a corneal epithelial wound healing stimulator, is a key regulator of EMT [46,47]. Additionally, *VIM* and *FN1* are typical mesenchymal cell markers, making them EMT-regulatory genes [48–50]. Our results supported the idea that AdMSC-EVs enhanced EMT in LECs. Moreover, increased cell size with the EV-treatment was also consistent with the morphological changes following EMT events. The observed maintenance of stemness and EMT by AdMSC-EVs underscores their potential as a therapeutic tool for regenerative medicine. However, it is important to consider the complex balance between beneficial and unbeneficial pathway activation by EVs, and in doing so, choosing a concentration whereby the beneficial pathway effects outweigh and/or limit the unbeneficial effects. Although a higher concentration of EVs could potentially increase desirable effects, surpassing an efficiency threshold may then activate regulators of negative pathways, relating to stemness and EMT; potentially limiting high-dose EV administration therapeutic effects due to saturation.

To determine the active factors in AdMSC-EVs, miRNAs were investigated, because they serve as post-transcriptional gene expression regulators [51]. The 3' UTR of SMAD Family Member 7 (*SMAD7*) gene is directly targeted by *miR-25-3p*. SMAD7 protein negatively regulates TGF-β signaling by binding to TGF-β type I receptor [52,53]. According to this, *miR-25-3p* may promote TGF-β-induced EMT in LECs *via* downregulation of *SMAD7*. Previous investigations have reported that Kruppel-like factor 6 (*KLF6*) is likely to be the target gene of *miR-191-5p* in LECs, and KLF6 can downregulate mesenchymal markers, such as *SNAIL*, *SLUG* and *VIM*, in several cell types [54,55]. Thus, it's possible that *miR-191-5p* may promote EMT of LECs by inhibiting *KLF6*. It's been confirmed that exosome-transmitted *miR-335-5p* facilitates EMT of colorectal cancer cells *via* direct targeting of the specific region of the 3' UTR of RAS p21 protein activator 1 (*RASA1*). RASA1 is an inhibitor of the Ras signaling pathway, therefore, the decreased expression of RASA1 protein *via miR-335-5p* upregulates Ras protein expression, inducing EMT, which causes elevated expression of the mesenchymal marker VIM and suppressed expression of the epithelial marker E-cadherin [56]. As the precise function of these miRNAs can vary depending on the specific cell type, surrounding environment, and targets which they interact with in the cells, as with other miRNAs [57],

according to the present study, *miR-25*, *miR-191*, and *miR-335* enhanced the colony formation ability and EMT of LECs. Interestingly, even though the collective effects of the miRNAs on colony formation ability of LECs were similar to those of the individual miRNA, the mechanisms between them might be different. When more kinds of miRNAs were involved, the interactions between these molecules and cells may become more complex.

When LECs and MMC-3T3 feeder cells were co-cultured, AdMSC-EVs were uptaken by both cell types. This implied that the improved CFE of LECs may be partly attributed to the influence of AdMSC-EVs on MMC-3T3 feeder cells. Interactions between LECs and corneal stromal fibroblasts are key factors to corneal functioning *in vivo* [58,59]. Bidirectional communication occurs as a specific and robust process during normal development and homeostasis, and when undergoing wound healing [58,59]. Since NIH 3T3 is a fibroblast cell line, we examined the effects of EV-treated MMC-3T3 feeder cells on LECs (S6 Fig). Using AdMSC-EV-pretreated MMC-3T3 cells, the AdMSC-EV-induced changes in MMC-3T3 cells were found to enhance the CFE of LECs in a dose-dependent manner. These results indicated that AdMSC-EVs can influence other supporting cells of LECs in a multicellular environment, whereby AdMSC-EVs may indirectly enhance the colony formation of LECs by activating MMC-3T3 feeder cells. Furthermore, the transfection of miRNAs into MMC-3T3 cells revealed that *miR-25*, *miR-191*, and *miR-335* were also potential molecules for MMC-3T3 cells to promote the colony formation ability of LECs. In addition, *miR-130*, which did not affect LECs, may indirectly alter the activities of LECs by affecting MMC-3T3 cells. These findings offer new insights into the potential role of AdMSC-EVs in the treatment of corneal epithelial diseases and reveal new targets for drug discovery.

The present study indicated that AdMSC-EVs enhanced the colony formation ability and EMT of LECs. It's been reported that cells undergoing EMT behave in many respects similarly to stem cells from both normal and neoplastic tissues [60]. Cells in hybrid EMT states where epithelial and mesenchymal markers are co-exhibited, are more likely to show stem cell characteristics compared to their fully epithelial and fully mesenchymal counterparts. And in most epithelial tissues, EMT activation is required to achieve a stem cell state [61]. Recent studies have suggested that EMT can activate well-established stem cell regulators of signaling pathways that directly promote stem cell properties [62,63]. Thus, in our study, the EMT induced by AdMSC-EVs could directly contribute to the enhanced colony formation ability of LECs.

Although functional miRNAs within AdMSC-EVs were identified, further investigation is required to clarify the mechanisms of action of these miRNAs in LECs. Indeed, more evidence, such as from *in vivo* studies, is required before the application of AdMSC-EVs in clinical therapies for corneal diseases.

## Conclusions

In summary, our study highlighted that AdMSC-EVs could exert a positive influence on both the colony formation and EMT of LECs through miRNAs. Notably, *miR-25*, *miR-191*, and *miR-335* were the most effective of highly expressed contributors to these effects.

## Materials and methods

### Ethics statement

Corneal limbus from human cadaveric donors used as cell sources were purchased for research specimens from the eye bank CorneaGen Inc. (Seattle, WA, USA) and were handled in accordance with the tenets of the Declaration of Helsinki.

## Cell culture

Human AdMSCs were purchased from PromoCell (Heidelberg, Germany) and grown in Cellartis MSC Xeno-Free Culture Medium (Cellartis; Takara Bio, Shiga, Japan). Cells were seeded into T75 cell culture flasks and digested with Accutase (Nacalai Tesque, Kyoto, Japan) for 10 min when 80–90% confluent. Passage 7 cells were used to isolate AdMSC-EVs at a seeding density of 2,500–3,000 cells/cm$^2$ in T75 flasks.

LECs were isolated from human cadaver corneas using an established method [64]. LECs were cultured in corneal epithelium maintenance medium (CEM; Dulbecco's modified Eagle's medium [DMEM]/F12; Thermo Fisher Scientific, Waltham, MA, USA), containing 2% B27 supplement (Thermo Fisher Scientific), 10 ng/ml KGF (FUJIFILM Wako Pure Chemical Corp., Osaka, Japan), and 10 μM Y-27632 (FUJIFILM Wako Pure Chemical Corp.). After expanding to 90–100% confluence in T75 flasks, the cells were digested with TrypLE Express (Thermo Fisher Scientific) for 10 min, and passage 2 cells were used directly in the following experiments.

## Isolation of EVs from CM

AdMSC-EVs were isolated by differential ultracentrifugation [65]. Briefly, after 2 d of culture, the AdMSC supernatant (passage 7) was decanted, and the cells were rinsed with phosphate-buffered saline (PBS; FUJIFILM Wako Pure Chemical Corp.). Fresh culture medium was added, and after 2 d of culture, the CM was collected. To eliminate cells, harvested supernatants underwent sequential centrifugation at 300 × g for 10 min at room temperature and 2,000 × g for 10 min at 4°C. Thereafter, the supernatant was centrifuged at 10,000 × g for 30 min at 4°C to remove cellular debris and macroparticles. Afterwards, ultracentrifugation was performed in an angle rotor (P50AT2; Hitachi Himac CP80WX preparative ultracentrifuge, Eppendorf Himac Technologies Co., Ltd, Ibaraki, Japan) at 100,000 × g for 70 min at 4°C. The pellet was resuspended in PBS, further undergoing ultracentrifugation at 100,000 × g for 70 min at 4°C. The final pellet was resuspended in PBS and preserved at 4°C for no more than 72 h.

## Quantification of EVs

Employing the Pierce™ BCA Protein Assay Kit (Thermo Fisher Scientific), protein concentrations in the EV samples were assessed *via* absorbance measurement at 570 nm using a plate reader (ARVO X4; PerkinElmer; Waltham, MA, USA).

## Scanning electron microscopy (SEM)

The isolated EV samples were processed as previously reported with minor modifications [66]. After fixation in 3.7% glutaraldehyde (Nacalai Tesque) for 15 min, the EVs were washed twice with PBS and collected on membrane filters, followed by dehydration with an ethanol concentration series (40%, 60%, 80%, 95%, and 100%) for 5 min. Finally, the samples were immersed in butyl alcohol for 5 min. The EV samples on membrane filters were dried using a freeze-drying device (JFD-320; JEOL Ltd., Tokyo, Japan). Membrane filters were adhered to the sampling stage using carbon tape. EV samples on the sampling stage underwent platinum coating using an autofine coater (FCL8 1600; JEOL) for 30 s and observed using SEM (JSM-6510LA; JEOL) at 20 kV.

## Western blot analysis

EV samples and AdMSC lysates (4 μg) were incubated at 95 °C for 5 min. After loading on NuPAGE 4–12% Bis-Tris gels (Thermo Fisher Scientific), the samples were transferred to polyvinylidene fluoride membranes (GE Healthcare, Chicago, IL, USA). Membranes were

blocked with 5% skim milk (FUJIFILM Wako Pure Chemical Corp.) in Tris-buffered saline with Tween-20 (TBST; Takara Bio) and adjusted to pH 7.6 for 30 min at room temperature. Membranes were then incubated with primary antibodies against CD63 (1:250, 10628D, Thermo Fisher Scientific), CD81 (1:250, 10630D, Thermo Fisher Scientific), TSG101 (1:500, HPA006161, Sigma, St. Louis, MO, USA), or calnexin (1:1000, #2679, Cell Signaling Technology, Danvers, MA, USA) at 4°C overnight, and next washed three times with TBST, and incubated with HRP-conjugated secondary antibodies (1:5,000; GE Healthcare) at room temperature for 1 h. Proteins were detected using ECL Select Western Blotting Detection Reagent (GE Healthcare) and visualised using a Molecular Imager ChemiDoc XRS+ system (Bio-Rad Laboratories, Hercules, CA, USA).

## miRNA transfection of human corneal epithelial cells

For colony formation assay, transfected LECs were seeded into a 24-well plate at a density of 25,000 cells/well and grown in CEM for 24 h. Subsequently, the medium was removed and the cells were transfected with either 10 nM individual/mixture miR mimics (*hsa-miR-25-3p*, *hsa-miR-126-5p*, *hsa-miR-130a-3p*, *hsa-miR-191-5p*, *hsa-miR-223-3p,* and *hsa-miR-335-5p*; Thermo Fisher Scientific; Table 2) or negative control (4464058, Thermo Fisher Scientific), in 1.25 μl of Lipofectamine™ RNAiMAX Transfection Reagent (Thermo Fisher Scientific) and 100 μl Opti-MEM® I Reduced Serum Medium (Thermo Fisher Scientific). After 24 h of incubation, the treated LECs were washed twice with PBS and digested with TrypLE Express. The collected LECs were immediately seeded onto MMC-3T3 feeder cells.

For quantitative real-time reverse transcription PCR, LECs were introduced into a 24-well plate at a density of 13,500 cells/well and cultivated in CEM for 24 h. Following the same transfection steps above and cultivation for 24 h, transfected LECs were rinsed once with PBS and cultured in CEM for 2 d, following which whole-cell RNA was extracted using Sepasol-RNA I Super G (Nacalai Tesque).

## miRNA transfection of MMC-3T3 cells

MMC-3T3 cells were seeded in 24-well plates at a density of 150,000 cells/well and cultured in Dulbecco's Modified Eagle Medium (DMEM; Thermo Fisher Scientific) supplemented with 10% fetal bovine serum (FBS; Thermo Fisher Scientific, Ref. 12483–020), for 24 h. MMC-3T3 cells were transfected using the same procedure used for LECs. After 24 h of culture in an antibiotic-free medium, the cells were washed twice with PBS, harvested by digestion with trypsin-EDTA (Thermo Fisher Scientific), and immediately used for subsequent experiments.

## Colony formation assay

MMC-3T3 cells were seeded as feeder layers in 12-well plates at a density of 18,000 cells/cm$^2$ and cultured in DMEM supplemented with 10% FBS for 2 d. LECs were seeded onto feeder layers at a density of 500 cells/well and fed with keratinocyte culture medium (KCM) [58].

**Table 2. Details of miRNA mimics used for cell transfection.**

| miRNA mimics | sequences |
|---|---|
| hsa-miR-25-3p | 5'-CAUUGCACUUGUCUCGGUCUGA-3' |
| hsa-miR-126-5p | 5'-CAUUAUUACUUUUGGUACGCG-3' |
| hsa-miR-130a-3p | 5'-CAGUGCAAUGUUAAAAGGGCAU-3' |
| hsa-miR-191-5p | 5'-CAACGGAAUCCCAAAAGCAGCUG-3' |
| hsa-miR-223-3p | 5'-UGUCAGUUUGUCAAAUACCCCA-3' |
| hsa-miR-335-5p | 5'-UCAAGAGCAAUAACGAAAAAUGU-3' |

The control groups received PBS, and EV-treated groups received protein concentrations of 30 μg/ml or 120 μg/ml. The medium was refreshed after 3 d, and after 6 d of cultivation, colonies were fixed with 4% paraformaldehyde (PFA; FUJIFILM Wako Pure Chemical Corp.) for 15 min, followed by staining with rhodamine B (FUJIFILM Wako Pure Chemical Corp.) overnight at room temperature. CFE was calculated by counting colonies larger than 0.08 mm² in size using the ImageJ software (version 1.53, National Institutes of Health, Bethesda, MD, USA).

### Colony formation assay of LECs pretreated with AdMSC-EVs

LECs were seeded in 24-well plates at 6,000 cells/well. Following 24 h culture in CEM, LECs were treated with 30 μg/ml or 120 μg/ml AdMSC-EVs or vehicle control (PBS), for 3 d. Subsequently, the cells were harvested by digestion with TrypLE Express and seeded onto MMC-3T3 feeder layers at 1,000 cells/well. After cultivation in KCM for 9 d, the colonies were fixed and stained as described in the section of Colony Formation Assay.

### Colony formation assay using the transfected LECs

Transfected LECs were seeded onto feeder layers at 1,000 cells/well and grown in KCM. After 6 d of cultivation, the colonies were fixed, stained, and analyzed as described in the section of Colony Formation Assay.

### Colony formation assay using the MMC-3T3 cells pretreated with AdMSC-EVs

MMC-3T3 cells were seeded as feeder layers onto 12-well plates at a density of 18,000 cells/cm² and treated with 30 μg/ml or 120 μg/ml AdMSC-EVs or vehicle control (PBS) for 2 d. LECs were seeded onto the feeder layers at a density of 500 cells/well. After 6 d of cultivation, the colonies were fixed, stained, and analyzed as described in the section of Colony Formation Assay.

### Colony formation assay using the transfected MMC-3T3 cells

Transfected MMC-3T3 feeder cells were cultivated in 12-well plates at a density of 18,000 cells/cm² for 2 d, and LECs were seeded at 1,000 cells/well. After 6 d of cultivation, the colonies were fixed, stained, and analyzed as described in the section of Colony Formation Assay.

### EV uptake assay

The ExoSparkler Exosome Membrane Labeling Kit-Red (Dojindo Laboratories, Kumamoto, Japan) was used according to the manufacturer's instructions. Briefly, cells were washed with PBS and refreshed with culture medium including 50 μl of labelled-EV solution or an equal volume of PBS. After 24 h of incubation, the medium was replaced, and images were captured using a fluorescence microscope (Axio Observer D1, Zeiss, Oberkochen, Germany).

### Scratch assay

LECs were seeded in 12-well plates at 300,000 cells/well and cultured in CEM for approximately 24 h until confluent. LEC monolayers were treated with 10 μg/ml mitomycin C (FUJIFILM Wako Pure Chemical Corp.) for 2 h and subsequently scratched using a p200 pipette tip. After three washes with PBS to remove debris, the cells were cultured for 24 h in CEM containing either AdMSC-EVs (30 μg/ml or 120 μg/ml) or an equal volume of PBS. Images at the same positions within each well were captured using EVOS® FL Auto Imaging System (Thermo Fisher Scientific), at time points immediately after scratching and 24 h later, and were analysed using ImageJ software.

## AlamarBlue assay

LECs were seeded in 24-well plates at a density of 4,500 cells/well. After 24 h culture in CEM, cells were washed twice with PBS, and treated either with 30 μg/ml or 120 μg/ml AdMSC-EVs, or an equal volume of PBS. After 3 d, cells were washed with PBS and incubated in CEM containing 10% AlamarBlue Cell Viability Reagent (Thermo Fisher Scientific). After 2 h, 100 μl of CEM-AlamarBlue mix was retrieved from each sample well and transferred to a new 96-well plate. Absorbance was measured at 550 nm and 590 nm using an ARVO X4 plate reader.

## RNA-Seq analysis

RNA-Seq analysis of LECs and AdMSC-EVs was performed by Rhelixa, Inc. (Tokyo, Japan), and Macrogen (Tokyo, Japan), respectively. Briefly, Sepasol-RNA I Super G (Nacalai Tesque) was used to extract total RNA from LECs and the miRNeasy Mini Kit was used for the RNA extraction from AdMSC-EVs. For LECs, a cDNA library was established using the NEBNext Ultra Ⅱ RNA Library Prep Kit for Illumina, and sequencing was performed on a NovaSeq 6000 platform (Illumina, San Diego, CA, USA). For AdMSC-EVs, a cDNA library was constructed using a SMARTER smRNA-Seq for Illumina Kit, and sequencing was performed on a HiSeq 2500 platform (Illumina). The sequencing quality was validated using FastQC v.0.11.7, followed by read trimming using Trimmomatic v.0.38. Sequenced reads were mapped onto the human hg38 reference genome using HISAT2 v.2.1.0. Quantification of raw reads mapped to exon regions was performed using FeatureCounts v.1.6.3. The DESeq2 package of R (version 4.2.2) was used to identify DEGs between samples, and the enrichment tree was exported from integrated Differential Expression and Pathway analysis (iDEP version 0.96).

## Cell morphological analysis

Cell segmentation for phase-contrast images was conducted using cellpose website (Cellpose version 1.0; https://www.cellpose.org) [67]. Measurements of cell area and cell circularity were operated using ImageJ software. The specific procedure was as follows: (1) Open the image of interest in ImageJ. (2) Convert to 8-bit image. (3) Image>Adjust>Threshold>Auto>Apply. (4) Select "wand (tracing) tool". Click the smallest cell to measure the area. (5) Analyze>Analyze particles. Set the minimum size.

## Immunostaining

Cell samples were fixed with 4% PFA in 24-well plates for 15 min, followed by 4–5 rinses with Tris-buffered-saline (TBS; Takara Bio). Nonspecific epitopes were blocked with TBS containing 5% normal donkey serum (NDS; Jackson ImmunoResearch, Bar Harbor, ME, USA) and 0.3% TritonX-100 (Sigma-Aldrich, St. Louis, MO, USA) for 30 min. Primary antibodies were diluted in TBS containing 1% NDS and 0.3% TritonX-100. Cells were incubated with mouse anti-TP63 (Abcam, Cambridge, UK; ab735, 1:100), rabbit anti-KRT12 (Abcam, ab185627, 1:1,000), rabbit anti-VIM (Abcam, ab92547, 1:200), or mouse anti-FN1 (Santa Cruz, Dallas, TX, USA; sc-271098, 1:120) antibodies overnight at 4°C. Cells were washed 4–5 times with TBS and incubated with Alexa-Fluor-488-conjugated anti-mouse (A21202, Thermo Fisher Scientific) or anti-rabbit (A21206, Thermo Fisher Scientific) secondary antibodies diluted 1:200 in TBS containing 1% NDS and 0.3% TritonX-100. Nuclei were stained with Hoechst 33342 (B2261-1G, 1:100, Sigma Aldrich). After 1 h of incubation at room temperature, the cells were rinsed with TBS 4–5 times. Cells were visualized in TBS and images were acquired using a confocal laser scanning microscope (FLUOVIEW FV3000; Evident Corp., Tokyo, Japan). Exposure times were identical for all treatments.

**Table 3. List of quantitative reverse transcription-polymerase chain reaction primers.**

| Target | Species | Forward | Reverse |
|---|---|---|---|
| GAPDH | Human | GGAGCGAGATCCCTCCAAAAT | GGCTGTTGTCATACTTCTCATGG |
| deltaNp63 | Human | TACCTGGAAAACAATGCCCAGA | GCGCGTGGTCTGTGTTATAG |
| KRT12 | Human | AGGAATAAGATCATTTCAGCCAGC | GCAGGGCCAGTTCATTCTCA |
| VIM | Human | GCTTCAGAGAGAGGAAGCCG | AAGGTCAAGACGTGCCAGAG |
| FN1 | Human | CTTCTGGTCAGCAACCCAGT | TCTTGTCCTACATTCGGCGG |
| BMI1 | Human | CCGCTTGGCTCGCATTCATT | TACCCTCCACAAAGCACACAC |
| SNAIL2 | Human | GCCAAACTACAGCGAACTGG | GATGGGGCTGTATGCTCCTG |
| KRT3 | Human | CCAGGAGCGGGAACAGATCA | TGAGATGGAACTTGTGCCCTG |
| CDH2 | Human | AACAGCAACGACGGGTTAGT | CAGACACGGTTGCAGTTGAC |
| LAMA3 | Human | TGGGATGGCTGTGGATCTTT | ACCCTTTGCTGCTGTGAACT |
| LAMC2 | Human | ACACTCAACACATTAGACGGC | CTGTTGATCTGGGTCTTGGCTC |
| MMP9 | Human | TGACAGCGACAAGAAGTGGG | TTCAGGGCGAGGACCATAGA |
| COL4A1 | Human | GGGGAGCCTGGTGAGTTTTATTT | CCTTTCAATCCTACAGAACCCG |
| COL4A2 | Human | GGACAGACGAGACAACAGCA | GAGCTGGCATAACATTGGCG |
| WNT5A | Human | GCCCAGGTTGTAATTGAAGC | TGGCACAGTTTCTTCTGTCC |
| TPM1 | Human | CTTGAAGTCACTGGAGGCTCA | ACTGACCTCTCCGCAAACTC |

## qRT-PCR

Total RNA was extracted from LECs using Sepasol-RNA I Super G (Nacalai Tesque) following the manufacturer's protocol, and cDNA synthesis was conducted using the SuperScript III First-Strand Synthesis System (Thermo Fisher Scientific). qRT-PCR was performed using an ABI PRISM 7500 Fast Sequence Detection System (Thermo Fisher Scientific). The thermocycling program was performed with an initial cycle at 95°C for 20 s, followed by 45 cycles at 95°C for 3 s and 60°C for 30 s. The relative expression level was calculated using the $2^{-\triangle\triangle Ct}$ method. GAPDH was used as a reference gene. The detecting reagent used was THUNDER-BIRD™ NEXT SYBR® qPCR Mix (Toyobo Inc, Osaka, Japan), and all primer sequences were designed using Primer–BLAST Web browser (Table 3).

## Statistical analysis

All data presented are expressed as the mean and standard deviations. Statistical analyses were performed using R software (version 4.2.2). One-way analysis of variance, followed by Dunnett's multiple comparisons test was utilized for parametric data (Figs 2C, 2F, 2H, 3D, 4E, S2B, S2C, S3, S4C, S5B, and S6C), whereas Steel's multiple comparison test was conducted for multiple comparisons of nonparametric data (Fig 1E, 4C, S4B, and S6F). Results were considered statistically significant at $p < 0.05$.

## Supporting information

**S1 Fig. Western Blotting of AdMSC-EVs.** Original images of CD63 (30–60 kDa), CD81 (~25 kDa), TSG101 (~46 kDa) and calnexin (~78 kDa) analysed by western blotting. EVs_24h/72 h: EVs were isolated from the conditioned medium that were cultured for 24 h/72 h. AdMSCs and 293FT cells were used as positive controls. White dashed rectangles show the cropped part used in Fig 1C.
(PDF)

**S2 Fig. Uptake of AdMSC-EVs and EMT-related analysis of EV-treated LECs.** (A) Images of the extracellular vesicles from human adipose-derived mesenchymal stem cells (AdMSC-EVs) labelled with Mem Dye-Red during the colony formation assay in the phosphate-buffered saline (PBS) control group and EV-treated group. (B) Comparisons of expression levels of additional genes related to epithelial-mesenchymal transition (EMT) by RNA-Seq analysis between the control and EV-treated groups. The results are shown as the mean ± SD. $n$ = 3 biological replicates. (C) Cell morphological analysis of EV-treated LECs. The results are expressed as the mean ± SD. $n$ = 4 biological replicates. NS: not significant. *$p < 0.05$, **$p < 0.01$, and ***$p < 0.001$.
(PDF)

**S3 Fig. Gene expression levels of miRNA-transfected LECs.** Quantitative reverse transcription-polymerase chain reaction analysis of genes related to stemness (*BMI1*, *SNAIL2*), differentiation (*KRT3*) and epithelial-mesenchymal transition (CDH2, *LAMA3*, *LAMC2, MMP9, COL4A1, COL4A2, WNT5A, TPM1*), compared between the NC group and microRNA (miRNA)-transfected groups. The results are expressed as the mean ± SD. $n$ =8 biological replicates.
(PDF)

**S4 Fig. Effects of grouped miRNA mixtures on LECs.** (A) Images showing colonies of the transfected LECs. $n$ = 16 biological replicates. (B) The colony-forming efficiency of LECs from the NC group and miRNA mixture groups, presented as the mean ± standard deviation (SD). $n$ = 16 biological replicates. (C) Quantitative reverse transcription-polymerase chain reaction analysis of *TP63, KRT12, VIM*, and *FN1* between the NC and miRNA mixture groups. $n$ =7 biological replicates. NS: not significant. *$p < 0.05$, **$p < 0.01$, and ***$p < 0.001$. NC: negative control. Mix(3): mixture of *miR-25, miR-191* and *miR-335*. Mix(6): mixture of *miR-25, miR-126, miR-130, miR-191, miR-223* and *miR-335*.
(PDF)

**S5 Fig. Images of LECs after 72 h treatment and analysis of genes related to p38 MAPK signaling pathway in EV-treated LECs.** (A) Original and magnified images of LECs after 72 h in the control and EV-treated groups. Red rectangles in the original images show the magnified part. (B) Comparisons of expression levels of genes encoding p38α (MAPK14) and p38β (MAPK11) between the control and EV-treated groups from RNA-Seq data. The results are shown as the mean ± SD. $n$ = 3 biological replicates. NS: not significant. *$p < 0.05$, **$p < 0.01$, and ***$p < 0.001$. NC: negative control.
(PDF)

**S6 Fig. Indirect effects of AdMSC-EVs on LECs.** (A) Methodology of colony formation assay using MMC-3T3 feeder cells pretreated with AdMSC-EVs. (B) Images showing colonies of LECs. MMC-3T3 feeder cells were pretreated with either PBS (control) or AdMSC-EVs (EVs_30 μg/ml and EVs_120 μg/ml). $n$ = 6 biological replicates. (C) The colony-forming efficiency of LECs from the PBS (control) group and EV-treated groups, presented as the mean ± standard deviation (SD). $n$ = 6 biological replicates. (D) Schematic representation of the assessment of MMC-3T3 feeder cells transfected with miRNAs. (E) Images displaying colonies of LECs from the NC group and the miRNA-transfected groups. $n$ = 15 biological replicates. (F) The CFE of LECs from the NC group and miRNA-transfected groups presented as the mean ± SD. $n$ = 15 biological replicates. *$p < 0.05$, **$p < 0.01$, and ***$p < 0.001$. NC: negative control.
(PDF)

## Acknowledgments

We thank Y. Ishikawa and T. Katayama at Osaka University for their technical assistance.

## Author contributions

**Conceptualization:** Xiaoqin Li, Ryuhei Hayashi, Tsutomu Imaizumi.

**Data curation:** Xiaoqin Li.

**Formal analysis:** Xiaoqin Li.

**Funding acquisition:** Ryuhei Hayashi, Kohji Nishida.

**Investigation:** Xiaoqin Li, Yuji Kudo, Koichi Baba.

**Methodology:** Xiaoqin Li, Ryuhei Hayashi, Tsutomu Imaizumi, Koichi Baba.

**Project administration:** Ryuhei Hayashi, Kohji Nishida.

**Resources:** Ryuhei Hayashi, Koichi Baba, Kohji Nishida.

**Software:** Ryuhei Hayashi, Koichi Baba, Kohji Nishida.

**Supervision:** Ryuhei Hayashi, Tsutomu Imaizumi, Jodie Harrington, Hiroshi Takayanagi, Kohji Nishida.

**Validation:** Xiaoqin Li, Tsutomu Imaizumi, Yuji Kudo.

**Visualization:** Xiaoqin Li.

**Writing – original draft:** Xiaoqin Li.

**Writing – review & editing:** Ryuhei Hayashi, Tsutomu Imaizumi, Jodie Harrington, Hiroshi Takayanagi.

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
