## [Decision Letter · Decision Letter 0]

15 Oct 2024

PONE-D-24-38324Extracellular vesicles from adipose-derived mesenchymal stem cells promote colony formation ability and EMT of corneal limbal epithelial cellsPLOS ONE

Dear Dr. Hayashi,

Thank you for submitting your manuscript to PLOS ONE. After careful consideration, we feel that it has merit but does not fully meet PLOS ONE’s publication criteria as it currently stands. Therefore, we invite you to submit a revised version of the manuscript that addresses the points raised during the review process.

**ACADEMIC EDITOR: **

This paper makes a significant contribution to the field of corneal regeneration and stem cell therapy by exploring the use of AdMSC-EVs and their miRNAs. While there are areas that need further clarification and discussion, the overall experimental design is solid, and the findings provide a promising foundation for future studies in regenerative medicine. With additional in vivo validation and deeper mechanistic insights, this study could have a substantial impact on the development of EV-based therapies for corneal diseases.

We look forward to receiving your revised manuscript.

Kind regards,

Mohamed A. M. Ali, Ph.D.

Academic Editor

PLOS ONE

Journal requirements: When submitting your revision, we need you to address these additional requirements. 1. Please ensure that your manuscript meets PLOS ONE's style requirements, including those for file naming. The PLOS ONE style templates can be found at https://journals.plos.org/plosone/s/file?id=wjVg/PLOSOne_formatting_sample_main_body.pdf and https://journals.plos.org/plosone/s/file?id=ba62/PLOSOne_formatting_sample_title_authors_affiliations.pdf 2. PLOS ONE now requires that authors provide the original uncropped and unadjusted images underlying all blot or gel results reported in a submission’s figures or Supporting Information files. This policy and the journal’s other requirements for blot/gel reporting and figure preparation are described in detail at https://journals.plos.org/plosone/s/figures#loc-blot-and-gel-reporting-requirements and https://journals.plos.org/plosone/s/figures#loc-preparing-figures-from-image-files. When you submit your revised manuscript, please ensure that your figures adhere fully to these guidelines and provide the original underlying images for all blot or gel data reported in your submission. See the following link for instructions on providing the original image data: https://journals.plos.org/plosone/s/figures#loc-original-images-for-blots-and-gels.   In your cover letter, please note whether your blot/gel image data are in Supporting Information or posted at a public data repository, provide the repository URL if relevant, and provide specific details as to which raw blot/gel images, if any, are not available. Email us at plosone@plos.org if you have any questions. 3. PLOS requires an ORCID iD for the corresponding author in Editorial Manager on papers submitted after December 6th, 2016. Please ensure that you have an ORCID iD and that it is validated in Editorial Manager. To do this, go to ‘Update my Information’ (in the upper left-hand corner of the main menu), and click on the Fetch/Validate link next to the ORCID field. This will take you to the ORCID site and allow you to create a new iD or authenticate a pre-existing iD in Editorial Manager. 4. Thank you for stating the following financial disclosure:  [This work was supported in part by Fusion Oriented Research for Disruptive Science and Technology (JPMJFR210W) from the Japan Science and Technology Agency (JST), Osaka City Innovation Support Grant, and Grant-in-Aid for Scientific Research (23H03060 and 20H03842) from the Japan Society for the Promotion of Science (JSPS). X.L. was directly supported by Otsuka Toshimi and Hattori International Scholarships.].  Please state what role the funders took in the study.  If the funders had no role, please state: ""The funders had no role in study design, data collection and analysis, decision to publish, or preparation of the manuscript."" If this statement is not correct you must amend it as needed. Please include this amended Role of Funder statement in your cover letter; we will change the online submission form on your behalf.

Additional Editor Comments:

 **Expand Discussion on Mechanisms** : The role of miRNAs in mediating the observed effects is central to the study, but further elucidation of the exact signaling pathways impacted by miR-25, miR-191, and miR-335 would be valuable. Are these miRNAs directly involved in the regulation of known EMT or stemness-related pathways?

 **In Vivo Studies** : Consider including in vivo models in future work to validate the therapeutic potential of AdMSC-EVs in a clinical setting. This will enhance the translational relevance of the findings.

 **Clarify the Relationship between Proliferation and Colony Formation** : More explanation is needed to reconcile the seemingly paradoxical observation that EVs reduce proliferation while enhancing colony formation. Are there alternative explanations, such as altered cell migration or differentiation status?

 **Link EMT with Stemness** : If EMT is driving regenerative properties, the manuscript should make a stronger case for the connection between these two processes. Consider referencing more literature that links EMT with stem cell maintenance, especially in epithelial tissues.

Reviewers' comments:

Reviewer's Responses to Questions

**Comments to the Author**

1. Is the manuscript technically sound, and do the data support the conclusions?

Reviewer #1: Yes

Reviewer #2: Yes

2. Has the statistical analysis been performed appropriately and rigorously? 

Reviewer #1: Yes

Reviewer #2: Yes

3. Have the authors made all data underlying the findings in their manuscript fully available?

Reviewer #1: Yes

Reviewer #2: Yes

4. Is the manuscript presented in an intelligible fashion and written in standard English?

Reviewer #1: Yes

Reviewer #2: Yes

5. Review Comments to the Author

Reviewer #1: In methods the authors continuously referred “as previously described” but no reference is added.

Figure 1, EV 30ug/mL shows statistically significant data than the EV 120 ugl/mL. Authors need to clarify this, or need to have more control over experimental data, especially SD.

Figure 2, the data provided don’t support each other. Colony formation is increasing but proliferation is decreased. Colony formation and proliferation are directly related to each other, if colony is increasing its direct indication of proliferation.

Figure 4, the authors transfected the cell with miR individually, there must be a group where all the miR in cocktail are transfected and then investigate the desired effect.

Overall the manuscript show quality data, and nicely drawn conclusion. Authors are required to addressed the highlighted issues.

Reviewer #2: Authors have shown the effect of AdMSCs Evs on LEcs, the manuscript has shown interesting findings. But there are a few concerns with the manuscript that have to be addressed and revised.

1. Authors are suggested to mention the functional aspect of EVs diversely by briefly mentioning their role in cancer, immunology, etc. Authors are advised to cite these publications. PMID: 36713536, PMID: 36570199.

2. Authors are suggested to mention the results only in the result section and the figure legend should be included in the last.

3. The images of electron microscopy 1B, is of low quality.

The images of Figure 2 E are also suggested to be replaced by high-quality images.

4. Figure 4D, images are to be replaced with high quality, nothing is visible in the pdf.

5. The discussion should be shortened.

6. Supplementary figures legend should be added in supplementary files only.

6. PLOS authors have the option to publish the peer review history of their article (what does this mean? ). If published, this will include your full peer review and any attached files.

**Do you want your identity to be public for this peer review?** For information about this choice, including consent withdrawal, please see our Privacy Policy .

Reviewer #1: No

Reviewer #2: **Yes: ** Namrata Anand

---

## [Author Response · Author response to Decision Letter 1]

7 Feb 2025

Response to Comments

Dear Dr. Mohamed A. M. Ali and reviewers:

Thank you for your comments concerning our manuscript entitled “Extracellular vesicles from adipose-derived mesenchymal stem cells promote colony formation ability and EMT of corneal limbal epithelial cells” (manuscript number: PONE-D-24-38324). The comments were very helpful for revising and improving our paper. We have studied the comments carefully and have made corrections that we hope meet with approval. Revised portions are marked in red within the Revised Manuscript with Track Changes. The main corrections for the paper and responses to each comment are as follows:

Response to the editor’s comments:

1. Expand Discussion on Mechanisms: The role of miRNAs in mediating the observed effects is central to the study, but further elucidation of the exact signaling pathways impacted by miR-25, miR-191, and miR-335 would be valuable. Are these miRNAs directly involved in the regulation of known EMT or stemness-related pathways?

Author Response: Thank you for your suggestions. These miRNAs, we believe, did directly involve in the regulation of EMT-related pathways. MiR-25-3p promotes TGF-β-induced EMT in LECs by directly targeting the SMAD7 gene, thereby downregulating SMAD7 protein that inhibits TGF-β signaling. MiR-191-5p may enhance EMT of LECs by inhibiting KLF6, a negative regulator of mesenchymal markers. Additionally, miR-335-5p might induce EMT by upregulating Ras protein expression. Since it’s known that EMT can induce stemness, we think it’s likely that these miRNAs enhanced the stem cell properties by improving EMT. The above discussions are shown in line 304-316.

2. In Vivo Studies: Consider including in vivo models in future work to validate the therapeutic potential of AdMSC- EVs in a clinical setting. This will enhance the translational relevance of the findings.

Author Response: Thank you for your advice. We’ve done preliminary animal experiments using a rat model to examine the effect of AdMSC-EVs (as shown in the PDF file named "Animal experiments for the editor"). Unfortunately, the first preliminary experiments showed no significant difference between the EV-treated group and the PBS group in the wound healing model. EVs show the effects on corneal epithelial cells in vitro when there is sufficient uptake as shown in this study, therefore, this is probably due to insufficient uptake or concentration of EVs in vivo. It’s still a problem to determine the optimized concentration of AdMSC-EVs, as well as to improve the uptake of AdMSC-EVs for in vivo experiments, which will take more time to solve. In addition, this would suggest that in vivo studies require a huge amount of AdMSC-EVs compared to in vitro studies. Based on these facts, this study will focus on the effects and mechanism of action of AdMSC-EVs in vitro, and as suggested by the editor, we would like to conduct future research on their in vivo effects after resolving some of the issues regarding EV administration mentioned above.

3. Clarify the Relationship between Proliferation and Colony Formation: More explanation is needed to reconcile the seemingly paradoxical observation that EVs reduce proliferation while enhancing colony formation. Are there alternative explanations, such as altered cell migration or differentiation status?

Author Response: Thank you for the valuable suggestions. Various kinds of factors, including cell proliferation, migration, and stemness (differentiation), contribute to colony formation. In this study, we think that enhanced cell migration by AdMSC-EVs particularly increased colony formation. Previous studies (Ref [37-38]) indicated that augmented cell migration increases cell reaggregation, cell splitting and colony splitting, which can increase the number of formed colonies. Furthermore, enhanced stemness via EMT by AdMSC-EVs treatment would also increase colony formation. From the above, it is strongly suggested that the enhancement of cell migration and EMT by EVs comprehensively improved colony formation ability. The above discussion is described in lines 259-269, and 343-352.

4. Link EMT with Stemness: If EMT is driving regenerative properties, the manuscript should make a stronger case for the connection between these two processes. Consider referencing more literature that links EMT with stem cell maintenance, especially in epithelial tissues.

Author Response: Thank you for your helpful comment. We’ve added more sentences and related references (Ref [60-63]) in the discussion part about the connection between EMT and stemness (lines 343-352) according to the editor’s suggestion.

Response to the reviewers’ comments:

Reviewer #1:

1. In methods the authors continuously referred “as previously described” but no reference is added.

Author Response: We are truly sorry for the confusing usage of the phrase. What we meant here is that the methods are as mentioned in this manuscript, not described in previous reports. To avoid confusion, we’ve replaced these phrases with “as described in the section of Colony Formation Assay (lines 472, 476-477, 484-485, and 490-491).

2. Figure 1, EV 30ug/mL shows statistically significant data than the EV 120 ug/mL. Authors need to clarify this, or need to have more control over experimental data, especially SD.

Author Response: Thank you for your suggestions. Firstly, let us be clear that both EV 30 ug/mL and EV 120 ug/mL significantly increased the CFE of LECs compared to the control group, and there’s no statistically significant difference between EV 30 ug/mL and EV 120 ug/mL. This simply indicates that an EV concentration of 30ug/ml is sufficient and might saturate for its effects on colony formation. In addition, increasing the amount of AdMSC-EVs does not necessarily strengthen its effects because EVs contain diverse biologically active factors, including both positive and negative factors for colony formation or EMT. For instance, it's possible that at high concentrations of AdMSC-EVs, the effects of stemness and EMT may be weakened by increasing the amount of molecules that powerfully suppress EMT or stemness. We’ve added some sentences about the above discussion (lines 293-301).

3. Figure 2, the data provided don’t support each other. Colony formation is increasing but proliferation is decreased. Colony formation and proliferation are directly related to each other, if colony is increasing its direct indication of proliferation.

Author Response: Thank you for the reviewer’s valuable advice. We agree with the reviewer’s comment that proliferation largely contributes to colony formation. On the other hand, we would like to point out that cells with a high proliferative ability are not necessarily those with a high colony-forming ability. For example, generally, stem cells divide infrequently (so-called "slow cycling cells"), but they exhibit a high colony-forming efficiency. Contrarily, differentiated from stem cells, TA (transient amplifying) cells show higher proliferation than stem cells but low colony-forming efficiency because they rapidly proliferate and are exhausted earlier (Ref [35-36]). As mentioned before, many factors are involved in colony formation, such as cell proliferation, migration, and stemness. Thus, it is difficult to explain the increase or decrease in colony formation due to a single factor, but based on the results of this study, we believe that the enhancement of cell migration and stemness via EMT caused by EVs would contribute to colony formation. We have added these discussions to lines 259-269 and 343-352.

4. Figure 4, the authors transfected the cell with miR individually, there must be a group where all the miR in cocktail are transfected and then investigate the desired effect.

Author Response: Thank you for your helpful suggestions. We have conducted additional experiments (S5 Fig) using the LECs transfected with either the three most effective miRNA mimics (Mix(3)) or all the miRNA mimics (Mix(6)). The results showed that both transfected groups promoted the colony-forming efficiency compared to the negative control group. And the qRT-PCR revealed that Mix(3) and Mix(6) groups increased expression levels of EMT-related markers VIM and FN1. Additionally, deltaNp63 expression levels remained stable, while KRT12 expression was downregulated in the Mix(3) group and upregulated in the Mix(6) group. Overall, these results showed that the collective effects of these miRNAs on colony formation of LECs were similar to those of the individual miRNA. As mentioned, the mechanism of action of miRNAs in EVs is complex, and in order to maximize their effect, it is likely necessary to optimize the combination of their type, amount, and composition in detail. We have added some sentences about the additional experiment to lines 223-229 and 320-323.

Reviewer #2:

1. Authors are suggested to mention the functional aspect of EVs diversely by briefly mentioning their role in cancer, immunology, etc. Authors are advised to cite these publications. PMID: 36713536, PMID: 36570199.

Author Response: We appreciate your suggestion to cite the references you provided. We have read them and learned a lot. The recommended publications were cited in the revised manuscript [Ref 18-19] (lines 81-82).

2. Authors are suggested to mention the results only in the result section and the figure legend should be included in the last.

Author Response: Thank you for your mindful suggestions. As the reviewer suggested, we have revised the manuscript to mention experimental results only in the Results section whenever possible. Regarding the figure legends, we confirmed the submission guideline of PLOS ONE, and for figure captions, it says “Each figure caption should appear directly after the paragraph in which they are first cited” (MANUSCRIPT BODY FORMATTING GUIDELINES; https://journals.plos.org/plosone/s/submission-guidelines). Thus, following the guidelines, figure legends were left as it is.

3. The images of electron microscopy 1B, is of low quality. The images of Figure 2 E are also suggested to be replaced by high-quality images.

Author Response: Thanks for the reviewer’s suggestion. We think it is because the upload degraded the image quality when converted to a PDF file. To view the original high-quality TIF image files, you can download them from the links “Click here to access/download;xxx" in the PDF file.

4. Figure 4D, images are to be replaced with high quality, nothing is visible in the pdf. Author Response: Thanks for the reviewer's comment. As mentioned above, please download the original TIF image with higher quality. I believe you will then be able to see the details.

5. The discussion should be shortened.

Author Response: We appreciate the reviewer’s valuable advice. Based on the reviewer’s comments, we have shortened the discussion by removing some sentences related to the results. At the same time, we needed to add some discussion to address the editor’s and reviewer’s feedback. As a result, the overall word count of the revised manuscript has not changed significantly.

6. Supplementary figures legend should be added in supplementary files only.

Author Response: Thanks for the reviewer's careful reading of our manuscript. We confirmed the submission guideline of PLOS ONE, and for supporting information captions, it says “List Supporting Information captions at the end of the manuscript in a section titled ‘Supporting information’” (MANUSCRIPT BODY FORMATTING GUIDELINES; https://journals.plos.org/plosone/s/submission-guidelines). Thus, following this guideline, we left them as it is.

---

## [Decision Letter · Decision Letter 1]

10 Mar 2025

Extracellular vesicles from adipose-derived mesenchymal stem cells promote colony formation ability and EMT of corneal limbal epithelial cells

PONE-D-24-38324R1

Dear Dr. Hayashi,

We’re pleased to inform you that your manuscript has been judged scientifically suitable for publication and will be formally accepted for publication once it meets all outstanding technical requirements.

Kind regards,

Mohamed A. M. Ali, Ph.D.

Academic Editor

PLOS ONE

Reviewers' comments:

1. Have the authors have adequately addressed your comments raised in a previous round of review?

Reviewer #1: All comments have been addressed

Reviewer #2: All comments have been addressed

2. Is the manuscript technically sound, and do the data support the conclusions?

Reviewer #1: Yes

Reviewer #2: Yes

3. Has the statistical analysis been performed appropriately and rigorously? 

Reviewer #1: Yes

Reviewer #2: Yes

4. Have the authors made all data underlying the findings in their manuscript fully available?

Reviewer #1: Yes

Reviewer #2: Yes

5. Is the manuscript presented in an intelligible fashion and written in standard English?

Reviewer #1: Yes

Reviewer #2: Yes

6. Review Comments to the Author

Reviewer #1: The authors addressed the question raised during the review process; therefore, the manuscript is accepted in its current form.

Reviewer #2: The authors addressed the question raised during the review process; therefore, the manuscript is accepted in its current form.

7. PLOS authors have the option to publish the peer review history of their article (what does this mean? ). If published, this will include your full peer review and any attached files.

If you choose “no,” your identity will remain anonymous, but your review may still be made public.

Reviewer #1: No

Reviewer #2: **Yes**

---

## [Editor Report · Acceptance letter]

PONE-D-24-38324R1

PLOS ONE

Dear Dr. Hayashi,

I'm pleased to inform you that your manuscript has been deemed suitable for publication in PLOS ONE. Congratulations! Your manuscript is now being handed over to our production team.

Kind regards,

on behalf of

Professor Mohamed A. M. Ali

Academic Editor

PLOS ONE